# Elexacaftor-Tezacaftor-Ivacaftor as a Final Frontier in the Treatment of Cystic Fibrosis: Definition of the Clinical and Microbiological Implications in a Case-Control Study

**DOI:** 10.3390/ph15050606

**Published:** 2022-05-14

**Authors:** Giuseppe Migliorisi, Mirella Collura, Francesca Ficili, Tiziana Pensabene, Dafne Bongiorno, Antonina Collura, Francesca Di Bernardo, Stefania Stefani

**Affiliations:** 1Unit of Clinical Microbiology, ARNAS Civico-Di Cristina-Benfratelli, 90127 Palermo, Italy; gpp.miglio@gmail.com (G.M.); tiziana.pensabene@arnascivico.it (T.P.); antonina.collura@arnascivico.it (A.C.); francescadibernardo@gmail.com (F.D.B.); 2Cystic Fibrosis and Respiratory Pediatric Center, Children’s Hospital G. Di Cristina, ARNAS Civico-Di Cristina-Benfratelli, 90127 Palermo, Italy; mirella.collura@arnascivico.it (M.C.); francesca.ficili@arnascivico.it (F.F.); 3Department of Biomedical and Biotechnological Sciences, University of Catania, 95124 Catania, Italy; d.bongiorno@unict.it

**Keywords:** Cystic Fibrosis, elexacaftor-tezacaftor-ivacaftor, microbiology, airway colonization

## Abstract

The use of modulator drugs that target the Cystic Fibrosis transmembrane conductance regulator (CFTR) is the final frontier in the treatment of Cystic Fibrosis (CF), a genetic multiorgan disease. F508del is the most common mutation causing defective formation and function of CFTR. Elexacaftor-tezacaftor-ivacaftor is the first triple combination of CFTR modulators. Herein, we report on a one-year case-control study that involved 26 patients with at least one F508del mutation. Patients were assigned to two similar groups, and patients with the worse clinical condition received treatment with the triple combination therapy. The study aimed to define the clinical and especially microbiological implications of treatment administration. The treatment provided significant clinical benefits in terms of respiratory, pancreatic, and sweat function. After one year of therapy, airway infection rates decreased and pulmonary exacerbations were dramatically reduced. Finally, treated patients reported a surprising improvement in their quality of life. The use of triple combination therapy has become essential in most CF people carrying the F508del mutation. Although the clinical and instrumental benefits of treatment are thoroughly known, further investigations are needed to properly define its microbiological respiratory implications and establish the real advantage of life-long treatment with elexacaftor-tezacaftor-ivacaftor.

## 1. Introduction

Cystic Fibrosis (FC) is the most common life-threatening autosomal recessive disease in the Caucasian population, affecting about 50,000 people in Europe [1]. It is caused by mutations in the gene encoding for the *cystic fibrosis transmembrane conductance regulator* (CFTR). To date, more than 2000 variants are described for the *CFTR* gene, but F508del remains the most prevalent mutation, affecting approximately 85.3% of CF patients in Europe [2].

The CFTR protein is an epithelial anion channel involved in the transport of chloride and bicarbonate on the surface of cells, where it regulates salt and water balance. Any decrease in or absence of CFTR activity leads to multifaceted clinical manifestations [3]. Cystic fibrosis is a chronic and progressive clinical disorder that affects the pulmonary, gastrointestinal, pancreatic, and reproductive systems. Pulmonary disease represents the most problematic clinical issue in people with Cystic Fibrosis (pwCF) [4].

Notably, defects in the *CFTR* gene affect the most important regulator of airway surface liquid hydration. The impairment of mucociliary clearance is the leading cause of the progressive increase in the amount of mucus collected in the airways. The resulting pulmonary environment allows airway pathogens to proliferate and toxic neutrophil mediators to accumulate, causing a vicious cycle of airway infection and inflammation that leads to progressive lung parenchymal damage and bronchial destruction, known as bronchiectasis. Airways infections and respiratory failure are the primary cause of morbidity and early mortality in pwCF [5].

CFTR modulator drugs have recently become the final frontier in the treatment of CF. They improve or even restore expression, function, and stability of CFTR in the presence of specific mutations in distinct manners. Depending on their effects on CFTR mutations, they are classified into five main groups: potentiators, correctors, stabilizers, read-through agents, and amplifiers. To date, four CFTR modulators have been licensed for the treatment of pwCF carrying specific CFTR mutations [6,7]. Elexacaftor-tezacaftor-ivacaftor, marketed as Trikafta^®^ (FDA) or Kaftrio^®^ (EMA), is the first triple combination therapy containing two correctors and a potentiator of the channel [8]. In June 2021, it was approved in Italy for use in patients aged 12 years and older with one F508del mutation (F/any) in the *CFTR* gene [9]. In Italy, the triple combination therapy was allowed for compassionate use before this date. Herein, we report on a one-year case-control study of 26 patients enrolled at the Regional Reference Centre for Cystic Fibrosis in Palermo, Italy.

## 2. Results

Data were retrospectively extracted from the patients’ medical records and from each follow-up visit. Table 1 refers to control group patients, and Table 2 reports relevant clinical data of patients on triple combination therapy.

### 2.1. Clinical Results


*FEV*
_1_


The spirometric analyses show an increase of approximately 10–15% in ppFEV1 in all treated patients compared to the previous data collected with no therapy in place. On the other hand, no change was reported in this respect in control group patients, where ppFEV1 values were steady.

Radiological Findings

The computed tomography (CT) scan, performed on treated patients only, detected a reduction in pulmonary damage and bronchial destruction. All treated patients still had signs of structural changes in their respiratory tissue, including multiple bronchiectasis and scarring lesions associated with pulmonary fibrosis. Even though these structural changes were still visible, imaging data showed no further parenchymal damages. Notably, signs of air trapping and mucoid impaction appeared to be less evident in all patients (100%) after treatment. Furthermore, parenchymal lung nodules and signs of regional lymphadenopathy disappeared after the elexacaftor-tezacaftor-ivacaftor combination therapy.

Nutritional Status

Both groups of patients maintained approximately the same body mass index (BMI) values during the period of observation, with slight but significant increases in the BMI of treated patients.

Sweat Chloride Values

In the majority of the cases (77%), the results of sweat tests reported a decrease in sweat chloride values, showing a trend towards the functional recovery of the sweat glands.

CFQ-R Questionnaire

Administered to all treated patients, the CFQ-R questionnaire showed absolute changes in the scores collected post-therapy: all patients (100%) reported a score of 100, indicating an improvement in their quality of life.

### 2.2. Microbiological Results

In all of the study patients (100%), the microbiological data report continuous airway colonization/infection rates. In line with the above, all sputum samples collected in both groups show constantly positive results. A number of 120 strains were collected and divided into *Staphylococcus aureus* (55), *Pseudomonas aeruginosa* (38), *Aspergillus niger* (1), *Achromobacter xylosoxidans* (5), *Candida albicans* (5), *Candida freundii* (1), *Candida lusitaniae* (1), *Candida parapsilosis* (2), *Enterobacter cloacae* (1), *Escherichia coli* (3), *Klebsiella pneumoniae* (3), *Proteus mirabilis* (1), *Stenotrophomonas maltophilia* (2), and *Streptococcus pneumoniae* (2). The most prevalent pathogens in the airways are *S. aureus* and *P. aeruginosa*, which are the relevant bacteria regularly reported in the sputum samples collected. 

The main difference between the two groups of patients consists in the most prevalent bacterial species detected in the respiratory samples (Table 3 and Table 4). *P. aeruginosa* is the most common bacterium found in treated patients: it was often detectable in a context of polymicrobial airway colonization in association with other clinically relevant microorganisms in CF. On the contrary, *S. aureus* is the most prevalent bacterium isolated from the control group patients’ sputum samples. 

Appendix A contains the results obtained for the treated patients vs. the control patients group.

Despite the above microbiological data, the sputum samples collected from case group patients after treatment show a decreasing rate of microbial colonization, progressively resulting in negative respiratory samples following no detection of relevant pathogenic microorganisms. The pulmonary colonization rates of treated patients dramatically decreased after just one year of therapy, resulting in almost half (45.3%) of the sputum samples analyzed during the treatment period becoming negative.

### 2.3. Statistical Analysis

Finally, this study related the airway colonization rates and number of pulmonary exacerbations in each patient. The statistical analysis showed a statistically significant reduction (*p* < 0.05) in the number of pulmonary exacerbations after one year of combination therapy with elexacaftor-tezacaftor-ivacaftor in case group patients. A total of 77% of these patients reported no further hospitalizations. By contrast, the same statistical analysis does not show a significant reduction in pulmonary exacerbations for the control group. 

## 3. Discussion

In 1989, the discovery of the CFTR gene provided an adequate understanding of the structure, processing, and role of the CFTR protein in the healthy epithelial tissue, even though it is still necessary to deeply understand the implications of the CFTR protein abnormalities. However, this tremendous amount of information enabled us to understand how any defect in this anionic channel can lead to multiorgan disease. In the last few decades, international research has designed new molecules that have the power to modulate the defective CFTR channel based on specific mutations occurring in CF patients. Modulator drugs have thus become the most promising and newest therapy in the treatment of CF [10].

The latest therapeutic option is the triple combination of elexacaftor-tezacaftor-ivacaftor. This is the first triple combination of modulator drugs approved for the treatment of pwCF aged 12 years and older carrying at least one F508del mutation in the *CFTR* gene. Since F508del is the most prevalent mutation in pwCF worldwide, the triple combination therapy is currently the treatment option for most of these patients. The use of modulator drug therapy, in particular the triple combination therapy, has become essential in a disease characterized by chronic symptomatic therapies only [11]. Therefore, modulator drugs revolutionized the way of thinking about the management and treatment of pwCF [12].

The triple combination therapy results in significant clinical benefits that exceed any results reported with the previous modulator drugs used alone or in combination [13,14,15]. In line with this, the present study highlights how only one year of treatment with elexacaftor-tezacaftor-ivacaftor is sufficient to produce benefits that can be appreciated in several clinical and laboratory parameters. The design of this case-control study allowed us to define any changes in every selected parameter and compare their evolution in both groups of patients, whose only difference concerns therapy administration. 

The main limitation of this study lies in the small number of CF patients enrolled, but it should be considered that elexacaftor-tezacaftor-ivacaftor was formally approved in Italy only in June 2021. Before this date, triple combination therapy was only provided for compassionate use. For this reason, the present study made this treatment available only for a limited group of CF patients, i.e., those with worse clinical condition.

ppFEV1 values reflect a gradual improvement in the respiratory function in case group patients. A 10–15% increase was seen compared to the ppFEV1 values observed during the previous period without triple combination therapy. Post-treatment ppFEV1 values still mirror an unhealthy respiratory condition (ppFEV1 > 50–60%), but the increase in ppFEV1 results in the absence of severe pulmonary disease and critical airway obstruction. Nevertheless, the radiological signs of persistent bronchial obstruction remain even after triple combination therapy. The effects of obstinate inflammation are the most prevalent radiological findings collected from the chest CT scan of every treated patient. The structural changes in the parenchymal respiratory tissue and bronchial airways include permanent damages that not even modulator drugs can remove. Because of their permanent nature, these structural changes persist even after long-term treatment. They are attributable to the lifelong vicious cycle of airway infection and inflammation that usually affects the respiratory system of pwCF for many years [5,16,17]. 

All treated patients enrolled in this study were above 18 years of age and had severe pulmonary disease (ppFEV1 < 40%). For this very reason, all of them had already experienced a chronic phlogistic state responsible for the permanent structural changes in the pulmonary environment. Nevertheless, even though the elexacaftor-tezacaftor-ivacaftor triple combination therapy cannot remove these damages, it still can aid in avoiding further alterations that may result in increased morbidity and mortality rates in pwCF.

The triple combination therapy has the power to improve the CFTR protein activity in the whole pulmonary system. This reduces the obstruction and the high amount of mucus collected in the airways, leading to considerable improvements in the clinical pulmonary disease. The reduction in the mucus amount is key to a less persistent inflammation state. This makes the pulmonary tissue inappropriate for the proliferation of pathogenic microorganisms. The lower rate of airway microbial colonization is the leading cause of decreasing pulmonary exacerbations in treated patients.

To the best of our knowledge, there is limited scientific evidence about the implications of the elexacaftor-tezacaftor-ivacaftor therapy on lung microbial diversity. Although the clinical and instrumental benefits of treatment are thoroughly known, it was surprising to notice a gradual decrease in the number of positive sputum samples. Airway infections are the primary concern for the lifelong health of CF patients. Several airway infections can occur in pwCF since their first months of life. These are always responsible for the unavoidable decline in respiratory function. Therefore, triple combination therapy seems to be the perfect way to prevent incessant infection and inflammation in pwCF carrying the specific F508del mutation. Thus, our primary goal was to observe the decreasing rate of infections, even those sustained by microorganisms usually responsible for chronic airway infections [18].

After only one year of triple combination therapy, 45.3% of the sputum samples collected were negative. There are reports in the literature of the impact of modulator drugs on CF lung microbiology, but they all refer to previous modulators. Only ivacaftor proves a direct effect on the lung microbiota, whereas lumacaftor induces cellular production of damaging reactive oxygen species. Ivacaftor is known to include a quinoline ring in its molecular structure, so it has already been proven capable of reducing the growth of *S. aureus* and *P. aeruginosa* through the weak inhibition of bacterial DNA gyrase and topoisomerase IV [19,20]. Even the combination lumacaftor/ivacaftor suggests a moderate change in the lung microbiota [21]. Moreover, a recent study documented the chance of elexacaftor-tezacaftor-ivacaftor in shifting the microbiome and even metabolome in the CF lung [22]. Although triple combination therapy is a recent combination, its clinical use resulted in a possible reduction in the microbial colonization rate of the airway that we still need entirely to understand.

However, on this basis, it is reasonable to think that the action of triple combination therapy, restoring CFTR protein function, leads to a partial recovery of the respiratory function and to a lower microbial colonization rate at the same time. The three different combinations of modulators may probably have potent activity even on the pulmonary microbiota, preventing colonization and reducing infection rates. In addition, we actually do not know if, beside ivacaftor, any of the other modulator drugs (elexacaftor and tezacaftor) have a specific antimicrobial effect.

It is essential to notice how the total benefits of triple combination therapy produce an improvement in the general health of pwCF. The absolute change in the quality of life, as shown by the CFQ-R after many months of treatment, is highly remarkable. All treated patients (100%) report the highest score on CFQ-R, which is a distinctive sign of a general recovery in their health. The CFQ-R is the best validated and most widely used questionnaire in CF. It allows patients to self-report any symptoms or changes in any aspect of their own life or health after a new therapy, such as modulator drugs. For each of our treated patients, the CFQ-R score was calculated on a 0–100 scale, with higher score indicating better patient-reported outcomes (PROs) [23].

The triple combination therapy shows its effects even on nutritional status. After one year of treatment, all patients maintained or reached their ideal BMI. We need to consider that some of these patients have exocrine pancreatic insufficiency and need adequate pancreatic enzyme replacement therapy (PERT). Some scientific evidence shows that triple combination therapy may even restore pancreatic sufficiency, resulting in PERT being unnecessary [24,25]. Further investigations are needed to fully understand the correlation between modulator drugs and pancreatic function but, as for our study, the positive effect of modulators on the nutritional status of our patients is undeniable.

Triple combination therapy shows its most significant effects in sweat test results. This therapy seems to have a positive impact even on sweat gland function, resulting in a progressive trend towards normalized and physiological values in the majority of our case group patients. The lower chloride rate ensures better thermoregulation and improved capacity to maintain salt-water balance, providing an opportunity to practice any physical activity without any peculiar clinical complications [26].

## 4. Materials and Methods

A case-control study was performed in CF patients at the Regional Reference Centre for Cystic Fibrosis in Palermo, Italy, in 2020–2021. Twenty-six patients were enrolled and divided into two groups similar in age, gender, genotypes and clinical features (Table 5). The underlying genotypic characteristics were well known for each patient. The specific inclusion criteria for case group patients were: critical genetic condition (two F508del mutations-F/F-or one F508del mutation and one minimal function mutation-F/MF) and severe pulmonary disease (percentage of predicted FEV_1_, ppFEV1, <40%). Because of their worse clinical and genetic status, this group of patients received compassionate use of elexacaftor-tezacaftor-ivacaftor combination therapy. Inclusion criteria for the control group included: presence of at least one F508del mutation and mild pulmonary disease (ppFEV_1_ > 50–60%). None of the control group genotypes was eligible for any of the CFTR modulator drugs available at the time of the clinical study. For these reasons, this group of CF patients did not receive treatment.

In order to gather clinical data for each CF patient, the following clinical and laboratory parameters were performed at each follow-up visit: ppFEV1 by spirometry, body mass index (BMI) by nutritional evaluation, total number of pulmonary exacerbations and airway microbial colonization status by sputum culture analyses. Furthermore, because of their worse clinical condition, additional tests were performed in the case group patients: sweat test, computed tomography (CT) scan of the chest and Cystic Fibrosis Questionnaire-Revised (CFQ-R). Data were gathered during one year of administration (2020–2021) for both groups of patients. In addition, to evaluate any changes in the same parameters, we also retrospectively collected data from the previous year (2019–2020) without triple combination therapy.

The ppFEV1, BMI and sweat test values were recorded into two 6-month observation periods (T_0_, T_6_, T_12_) during the prospective and retrospective collection of data in treated patients, while, for control group patients, ppFEV1 and BMI values were reported as the best values recorded per year of observation. The absolute number of pulmonary exacerbations was obtained by adding all episodes occurring over the two periods of observation. Treated patients also filled out a CFQ-R questionnaire prior to starting and after completing one year of treatment: a score was obtained for each CFQ-R to evaluate any changes in patient-reported outcomes (PROs). Imaging data were collected from chest CT scans performed in case group patients both prior to starting and after completing one year of treatment. Lastly, sputum samples, collected at every follow-up visit, were analyzed following the current Italian guidelines on microbiological procedures for the processing of CF respiratory samples at the Microbiology and Virology Unit of Civico-Di Cristina-Benfratelli Hospital in Palermo, Italy [27,28].

The samples were inoculated into enriched and selective agar media after dilution, such as blood agar, MacConkey agar, Mueller-Hinton agar, Sabouraud agar and OFPBL (oxidation-fermentation-polymyxin-bacitracin-lactose) agar. The isolated microorganisms were identified by MALDI-TOF Bruker, while the susceptibility tests were performed using BD Phoenix or Microscan Walkaway.

Statistical analyses were also performed using the Fisher’s exact test. These data were used to compare qualitative changes between the two groups of CF patients: a *p*-value < 0.05 was considered significant. Each patient included in the study sample provided an informed consent for the study.

## 5. Conclusions

Triple combination therapy can profoundly modify the natural history of CF. Based on our findings, long-term treatment with elexacaftor-tezacaftor-ivacaftor can give rise to considerable clinical changes in pwCF, especially in CF pulmonary microbiology. An effective modulator therapy, such as the triple combination therapy, may reduce the need for antibiotics, avoiding an enormous selective pressure on the lung microbiota. For this reason, we hope that this study will lead to more comprehensive future research to clarify the interactions between triple combination therapy and microorganisms, as well as to explain how modulator drugs can mitigate the pulmonary microbiota.

The main future outlook includes the possibility of using the modulator therapy even in children < 12 years of age with the F508del mutation, whose efficacy was proven in a recent phase 3 clinical trial [29,30]. This evidence may lead to the early use of elexacaftor-tezacaftor-ivacaftor, which would have major implications on the pulmonary function of patients. Early administration may reduce airway infections, resulting in fewer structural changes, which may be the way to prevent rapid decline in pulmonary function and provide a better quality of life for longer.

## Figures and Tables

**Table 1 pharmaceuticals-15-00606-t001:** Clinical data of control group patients.

Patients	Best FEV1		Best BMI		Number of Pulmonary Exacerbations	
	2019/20	2020/21	2019/20	2020/21	2019/20	2020/21
1	56	60	27.6	28.9	0	1
2	81	86	20.1	20.1	0	0
3	78	62	24.2	25.0	0	1
4	65	67	21.7	20.8	0	0
5	108	104	24.3	22.1	0	0
6	65	68	24.4	24.8	0	1
7	61	67	17.9	17.7	1	0
8	66	74	20.7	20.7	0	1
9	116	117	25.1	25.2	0	0
10	64	72	22.4	23.2	1	1
11	63	66	22.3	23.4	0	0
12	106	74	24.3	24.1	0	1
13	72	86	24.9	25.4	0	0

Legend: borderline values; normal values.

**Table 2 pharmaceuticals-15-00606-t002:** Clinical data of case group patients.

Patients	Pre-Therapy FEV_1_			Post-Therapy FEV_1_			Pre-Therapy BMI			Post-Therapy BMI			Pre-Therapy Sweat Test			Post-Therapy Sweat Test			CFQR		Number of Pulmonary Exacerbations	
	T0	T6	T12	T0	T6	T12	T0	T6	T12	T0	T6	T12	T0	T6	T12	T0	T6	T12	Pre	Post	Pre	Post
1	28	35	37	49	57	60	17.2	17.5	17.4	17.1	17.1	17.5	82	84	78	86	84	78	33	100	4	0
2	29	37	24.6	32	45	58	26.2	26.2	26.9	27.1	27.2	27.2	109	90	86	86	44	35	34	100	2	0
3	30	28	30	32	45	44	20.2	19.4	20.1	19.2	24.1	24.8	119	100	90	88	66	65	36.6	100	2	1
4	23	25	21	18	51.7	52	19.1	19.6	19.4	21.6	24.1	23.0	109	105	98	103	46	36	22	100	5	0
5	22	27	22	26	48	60	19.3	20.1	19.5	19.5	21.5	22.3	67	75	72	67	36	30	72.2	100	9	0
6	27	25	28	29	38	45	17.6	18.0	17.7	20.3	22.9	22.1	98	95	96	98	74	100	49	100	3	1
7	45	37	45	37	49	45	25.1	25.3	25.2	25.3	25.2	25.4	60	68	65	107	60	38	33.3	100	1	0
8	20	18	15	18	24	45	15	15.1	14.8	14.8	18.2	19.5	109	119	98	104	73	36	44.4	100	7	0
9	21	15	17	17	36	42	19.1	19.2	18.2	20.7	22.4	22.7	78	86	90	95	50	48	61	100	3	0
10	33	31	33	25	42	46	21.4	19.7	20.8	21.4	25.4	26.6	109	98	109	109	67	38	77.8	100	12	0
11	25	27	25	27	45	56	17.7	18	17	18.2	18.5	18.6	93	82	82	93	34	37	78.9	100	2	2
12	25	27	29	22	38	43	19.2	20.0	19.4	20.1	24.2	23.1	130	118	101	124	37	38	44.4	100	1	0
13	22	25	28	30	45	50	7.3	7.8	8.1	7.3	14.5	15.4	128	108	115	138	46	48	22	100	10	0

Legend: pathological values; borderline values; normal values.

**Table 3 pharmaceuticals-15-00606-t003:** Microbial prevalence in case group patients.

Treated Patients	Before Treatment							After Treatment						
	Airway Colonization *p* < 0.05	*P. aeruginosa* Dry Colony	*P. aeruginosa* Mucoid Colony	*S. aureus*	*A. xylosoxidans*	Other Microorganisms	Clinical Exacerbations *p* < 0.05	Airway Colonization *p* < 0.05	*P. aeruginosa* Dry Colony	*P. aeruginosa* Mucoid Colony	*S. aureus*	*A. xylosoxidans*	Other Microorganisms	Clinical Exacerbations *p* < 0.05
1	9	11%		100%		67%	4	1			100%			0
2	6		100%			33.3%	2	2		100%	50%			0
3	6		100%			17%	2	7		100%	-			1
4	8		25%	87.5%	12.5%	12.5%	5	2			100%		50%	0
5	8	75%	12.5%	37.5%		12.5%	9	1	100%					0
6	9	22.2%			100%	44.4%	3	4				75%	50%	1
7	5	80%				40%	1	2			50%		50%	0
8	7		100%				7	1		100%	100%			0
9	8	87.5%	87.5%	62.5%			3	4	25%	100%	75%		50%	0
10	18	11.1%	5.5%	33.3%		100%	12	3	33.3%		100%		33.3%	0
11	6	67%				67%	2	1					100%	2
12	4	25%				75%	1	2			50%		100%	0
13	13	31%		85%	92.3%	46%	10	1	100%			100%	100%	0

**Table 4 pharmaceuticals-15-00606-t004:** Microbial prevalence in control group patients.

Control Patients	Period of Observation: 2019–2020							Period of Observation: 2020–2021						
	Airway Colonization	*P. aeruginosa* Dry Colony	*P. aeruginosa* Mucoid Colony	*S. aureus*	*A. xylosoxidans*	Other Microorganisms	Clinical Exacerbations	Airway Colonization	*P. aeruginosa* Dry Colony	*P. aeruginosa* Mucoid Colony	*S. aureus*	*A. xylosoxidans*	Other Microorganisms	Clinical Exacerbations
1	3		100%	33.3%		33.3%	0	5		80%	40%		40%	1
2	4			100%	50%	25%	0	2			50%	50%		0
3	3	33.3%		100%		33.3%	0	4			100%			1
4	3		100%	33.3%			0	2		100%				0
5	3			100%			0	3			100%		33.3%	0
6	3		100%	100%			0	6		17%	100%			1
7	5	40%		100%		20%	1	2	50%		100%			0
8	3		100%	33.3%		33.3%	0	5		100%	80%		20%	1
9	2			50%		50%	0	4			25%		75%	0
10	10	80%	100%	60%		10%	1	6	100%	83%	83%			1
11	2	100%		50%			0	5	20%	80%	80%		20%	0
12	5			100%		20%	0	2			100%		50%	1
13	5			100%			0	1			100%			0

**Table 5 pharmaceuticals-15-00606-t005:** General information about the two groups.

Patients Treated with Triple Combination Therapy			
Patients	Age	Gender	Genotype
1	25	M	DF508/2183 AA > G
2	50	M	DF508/DF508
3	48	M	DF508/del2 ins182
4	20	M	DF508/G542X
5	24	F	DF508/DF508
6	28	M	DF508/N1303K
7	35	M	DF508/2183 AA < G
8	21	F	DF508/DF508
9	23	F	DF508/L102R
10	23	F	DF508/DF508
11	29	F	DF508/DF508
12	43	F	DF508/E585X
13	18	F	DF508/del ex2
**Control Group Patients**			
**Patients**	**Age**	**Gender**	**Genotypes**
1	41	M	DF508/2789 + G > A
2	31	F	DF508/DF508
3	19	M	DF508/G542X
4	44	F	DF508/2183AA > G
5	19	F	DF508/D1152H
6	43	M	DF508/L558S
7	18	F	DF508/DF508
8	33	M	DF508/R1158X
9	30	M	DF508/DF508
10	22	F	DF508/G542X
11	40	M	DF508/2789 + 5G > A
12	36	F	DF508/2789 + 5G > A
13	34	M	DF508/G542X

## Data Availability

Data is contained within the article.

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
