# Peer review of "Elexacaftor-Tezacaftor-Ivacaftor as a Final Frontier in the Treatment of Cystic Fibrosis: Definition of the Clinical and Microbiological Implications in a Case-Control Study"

_pharmaceuticals, 2022, doi:10.3390/ph15050606_

Round 1

Reviewer 1 Report

The authors report a retrospective case-control study regarding the most advanced, highly effective modulator therapy (HEMT) in patients with CF. In agreement with multiple clinical reports, this study proves treatment with the elexacaftor-tezacaftor-ivacaftor (ETI) combination benefits patients with genotype carrying F508del variant at least on one allele. Undoubtedly, this treatment has revolutionized Cystic Fibrosis therapy and predictably will increase the life expectancy and quality of life for patients with CF. The manuscript demonstrates the effectiveness of the treatment by stating improved ppFEV1, BMI and decreasing bacterial colonization with reduced pulmonary exacerbation in the case group compared to the control one.

Critics:

  • Although tables with clinical parameters represent all the data, I believe it would be a valuable addition for individual box plots showing FEV1, BMI and sweat chloride changes in the control vs. case group.
  • Please rephrase the following sentence: “The pulmonary colonization rates of treated patients dramatically decreased after just one year of therapy, resulting in almost half (45.3%) of the sputum samples analyzed during the treatment period becoming negative.” These sputum samples became negative for only P. aeruginosa but revealed the presence of other microbial pathogens.
  • Would you please provide a more extensive description of microbial sputum analysis in the methods section? Was the microbial analysis performed by PCR or microbiological culture?
  • Please rephrase the following sentence:” In 1989, the discovery of the CFTR gene provided an adequate understanding of the structure, processing and role of the CFTR protein in the healthy epithelial tissue.” —Although the discovery of the CFTR gene was an important milestone in understanding the pathomechanism of CF disease, it did not provide any information on the CFTR protein abnormalities. Since the discovery of the gene, a tremendous effort has been made by the CF research community to discover the abnormalities, misfolding, instability of the CFTR protein as well as splicing abnormalities, instability of CFTR RNA. This research process is still an ongoing effort.
  • Please rephrase the following sentence: “The low amount of mucus is key to a less persistent inflammation state. This makes the pulmonary tissue inappropriate for the proliferation of pathogenic microorganisms.” — The low amount of mucus accumulation is the key. Mucus is necessary to remove bacteria and other pathogens, as well as small particulates, from the surface of the airway by the sweeping movement of airway epithelial cilia. Mucus is necessary to collect all these particles and microorganisms and carries them toward the pharynx. Although mucus overproduction is also described as a CF-specific phenomenon.
  • Please correct the following sentence: “After only one year of triple combination therapy, 45.3% of the sputum samples collected were negative.” (in discussion)— Please see “critics” point 2 regarding the problem.
  • Please add further sentences to these thoughts: “However, all of these studies only report a temporary and relative change in CF airway microbiology. Instead, a recent study documented the ability of elexacaftor-tezacaftor-ivacaftor to shift the microbiome and even metabolome in the CF lung [22].” —The triple combination therapy is relatively new. We do not know whether the treatment will decrease the microbial colonization of the airway in the long term.

Author Response

First of all, we are pleased to thank you for the review and the valuable observations above.

  • We have already reported the FEV1 and BMI changes of the two patients’ group in the tables mentioned above. Each table provides a comparison of the two clinical parameters inside two evaluation periods per group. Only the treated patients report the sweat test values because of their worst clinical condition. For this reason, the test was not performed on the control group patients because it wasn't included in their clinical management before the beginning of our study. In conclusion, we would not have any values regarding the sweat test in the control patients to compare in our retrospective analysis.
  • What we mentioned as "negative results" refers to the samples that resulted entirely negative at the microbiological analysis among all respiratory samples collected during the one-year observational period. At the same time, the table's section "after treatment period" reports the percentage of the respiratory microbial prevalence in the only samples that resulted in positive even though the therapy. In this case, the samples were considered positive regards the presence of P. aeruginosa or any other relevant microorganism to report. In conclusion, the percentage (45,3%) is indicative of the only negative samples for whatever relevant pathogens.
  • The microbial analysis was performed following the microbial cultural method that is well explained in the reference guidelines [27] provided with the manuscript. These guidelines are generally followed by each microbiological laboratory involved in cystic fibrosis, so we believe it is enough to state that we have followed these procedures well known in the field.
  • We modified the manuscript, see Lines 136-139
  • We modified the manuscript, see Lines 200-202
  • The answer is the same consideration reported on point 2.
  • We modified the manuscript, see Lines 226-236

Reviewer 2 Report

The proposed study is interesting, and the proposed conclusions are convenient from a clinical pharmacy viewpoint. 

Nevertheless, before publication, I recommended moderate revision. 

First, in my opinion, the proposed manuscript is too short to be published as a full research paper. Consequently, I suggested adding more exetensve discussion or alternatively changing the form to communication. 

My second suggestion, focusing on statistical analysis. In this form, it is not entirely reassuring and would need to be significantly extended to include the variance of the distributions. Indeed, the execution of heatmaps would introduce a lot of interesting information and improve the visual reception of the work.

Author Response

We are pleased to thank the reviewer for the review.

We tried to amplify the discussion making the paper suitable for publication as full article, see “discussion section lines 136-139; 200-202; 226-236”.

We are sorry, but we are not able to include other statistical analysis although we are convinced that these could improve the manuscript.

Reviewer 3 Report

The manuscript by Migliorisi et al reports a one-year case-control study with the elexacaftor/tezacaftor/ivacaftor (ETI) drug, that involved 26 patients with at least one F508del mutation. This study took place before the approval of ETI in Italy, thus only patients with a severe disease could be enrolled, under compassionate use. 

The study aims to define the clinical and microbiological implications of treatment administration. The treatment provided significant clinical benefits in terms of respiratory, pancreatic and sweat function. After one year of therapy, the authors report an airway infection rates decrease as well as a reduction in pulmonary exacerbations. Interestingly, the authors also report data about the implications of the ETI therapy on lung microbial diversity.

Treated patients also reported a marked improvement in their quality of life.

The control group and the case group patients have similar in age, gender, genotypes and "clinical features" (as stated by the authors) although there is a huge difference between the two groups, that is the severity of the disease. Considering that the study took place when ETI was available only under compassionate use, this difference is logical and expected. 

Author Response

We are pleased to thank the reviewer for the review and for understanding the meaning of our study.

Round 2

Reviewer 1 Report

I would like to thank the author for accepting the suggested modifications in my first review. My intention was to improve the quality and the accuracy of the manuscript with that review. The results presented in this manuscript are clearly representing the effectiveness of triple combination therapy regarding all clinical parameters.  Nevertheless, generating graphs to show the statistical significance of the results would help to visualize and make comprehend the significance of the study more deeply.

Respond to the answer about missing microbiological methodology:

The referenced link (27) is in Italian language. The author should not expect the reader understands the native language of the author. I understand the resource is written as a guide for microbiological procedures and possibly harmonized with other countries, but because this manuscript emphasizes on the microbiological findings it should have a clear methodology described in it.

I would like to ask the authors please include a methodology section on the microbiology part.

Author Response

Dear reviewer,

we appreciate your comments and revisions, and for this reason we add an appendix with some graph that can help the readers to better understand our paper (see lines 117-118 and appendix 1) and, also, a period in the text to explain the procedure adopted to identified the strains and a new reference (lines 322-325 and ref 28).
thank you